# Gender-Related Patterns of Emotion Regulation among Patients with Eating Disorders

**DOI:** 10.3390/jcm8020161

**Published:** 2019-02-01

**Authors:** Zaida Agüera, Georgios Paslakis, Lucero Munguía, Isabel Sánchez, Roser Granero, Jessica Sánchez-González, Trevor Steward, Susana Jiménez-Murcia, Fernando Fernández-Aranda

**Affiliations:** 1CIBER Fisiopatología Obesidad y Nutrición (CIBERobn), Instituto de Salud Carlos III, L’Hospitalet de Llobregat, 08907 Barcelona, Spain; Roser.Granero@uab.cat (R.G.); tsteward@bellvitgehospital.cat (T.S.); sjimenez@bellvitgehospital.cat (S.J.-M.); 2Department of Psychiatry, University Hospital of Bellvitge-IDIBELL, L’Hospitalet de Llobregat, 08907 Barcelona, Spain; paslakis@outlook.de (G.P.); isasanchez@bellvitgehospital.cat (I.S.); jsanchezg@bellvitgehospital.cat (J.S.-G.); 3Department of Public Health, Mental Health and Maternal-Child Nursing, School of Nursing, University of Barcelona, L’Hospitalet de Llobregat, 08907 Barcelona, Spain; 4Department of Clinical Sciences, School of Medicine and Health Sciences, University of Barcelona, L’Hospitalet de Llobregat, 08907 Barcelona, Spain; laarcreed_lm@hotmail.com; 5Departament de Psicobiologia i Metodologia de les Ciències de la Salut, Universitat Autònoma de Barcelona, 08193 Barcelona, Spain

**Keywords:** emotion regulation, males, eating disorders

## Abstract

Difficulties in emotion regulation (ER) are common in females with eating disorders (ED). However, no study to date has analyzed ER in males with ED. In the study at hand, we assessed ER in males with ED and compared results to both females with ED and healthy controls (HC). We also examined associations between ER difficulties, personality, and psychopathology. A total of 62 males with ED were compared with 656 females with ED, as well as 78 male and 286 female HC. ER was assessed by means of the Difficulties in Emotion Regulation Scale (DERS). We found that males and females with ED showed greater ER difficulties compared to HC. Pronounced general psychopathology was a shared factor associated with higher ER difficulties in both males and females with ED. However, whereas higher novelty seeking, higher cooperativeness, lower reward dependence, and lower self-directedness were related to higher ER difficulties in females with ED, lower persistence was associated with ER difficulties in males with ED. In sum, males and females with ED show similar ER difficulties, yet they are distinct in how ER deficits relate to specific personality traits. Research on strategies promoting ER in the treatment of males with ED is warranted.

## 1. Introduction

Emotion regulation (ER) is defined as the sum of techniques applied to manage the variety, intensity, and duration of emotions [1]. Such strategies range from the putatively less adaptive, such as dissociation, avoidance, or suppression, to the supposedly more adaptive, e.g., cognitive reappraisal or problem-solving. Difficulties in ER are a transdiagnostic feature among multiple mental disorders and may explain high comorbidity rates (e.g., with anxiety, depression, or borderline personality disorder) [2]. Accordingly, ER is proposed as a transdiagnostic target for treatment [3].

ER allows one to cope with aversive emotions, is a core feature of self-regulation, and has a profound influence on food intake behaviors [4]. Difficulties in ER are present across all types of eating disorders (ED) [5,6,7,8,9]. In some studies, anorexia nervosa (AN) and bulimia nervosa (BN) do not seem to significantly differ with regard to most domains of ER. Patients with binge-eating disorder (BED) show less severe ER difficulties than patients with AN or BN [5,10], although there are also studies claiming that patients with binge-eating episodes (BED, BN, and AN/binge-eating purging subtype) present more ER difficulties compared to patients with AN/restrictive subtype [11] and others have shown no differences across ED types [12]. Nevertheless, ED are associated with other behaviors linked to ER difficulties, such as substance abuse and self-harm [9,13,14,15].

In addition, it is unclear to which degree difficulties in ER in ED may be seen as etiopathogenetic/vulnerability or as a maintenance factor contributing to the perpetuation of the disorder. In AN, starvation and low body weight reduce the susceptibility for emotional stimuli in the short-term and are thought to serve as dysfunctional strategies to regulate aversive emotions [16,17]. Patients suffering from AN are known to have difficulties in identifying emotional states in themselves and in others (i.e., alexithymia) and may, in part, be reversed parallel to weight gain during the course of treatment [18]. This is of clinical relevance, since difficulties in identifying emotions in others are associated with difficulties in one’s own ER skills [19,20]. Relatedly, the interrelation between ER and binge-eating behavior postulated in different models. According to the affect regulation theory [21], binge-eating episodes in BN are used to relieve states of negative affect. In their meta-analysis, Haedt-Matt and Keel [22] showed that negative affect immediately before an episode of binge-eating is higher than a day’s average affective content and higher than the dominant affect immediately prior to an unobtrusive eating behavior. In opposition to the affect regulation theory, the aversive emotional state does not resolve immediately after the binge-eating episode, but after an apparent delay of several hours [22]. Following a binge episode, compensatory behavior in BN may prevent a further increase in negative affect. In addition, a prior study analyzing ER in female ED patients before and after treatment found that emotional dysregulation can be modified as an effect of symptomatological ED improvement [23]. With these controversial results in mind, the question of whether the emotional dysregulation is a vulnerability factor for ED, a factor that maintains and worsens with the ED or both, is still open. At present, several manualized therapies for ED focusing on ER have been published [24,25].

Nevertheless, as in the vast majority of ED studies, females are overrepresented in studies on ER in ED [7,8], and males with ED are not researched as a whole. Although females greatly outnumber males with respect to diagnosed ED, it stands to reason that ER could also play a role in eating pathology in males, as it does in females.

With regard to gender-related differences in ER, evidence is scarce and mostly derived from studies in nonclinical community samples of males and females. In the study by Hayaki and Free [26], difficulties in ER predicted disordered eating in both male and female undergraduate students. Whereas some studies have shown no global differences between genders in nonclinical cohorts [27], others have shown gender-specific affective responses to high-calorie visual cues [28]. Significantly higher levels of rumination have also been identified in females, which, as an ER strategy, mediated the relationship between gender and disordered eating [29]. Difficulties in ER were identified as important determinants of body dissatisfaction and disordered eating in a study with only undergraduate males [30].

In a recent study in a cohort suffering from ED, difficulties in ER were found to be more strongly associated with cognitively oriented ED symptoms than with behavioral symptoms, such as binge eating, purging, driven exercise, non-suicidal self-injury, or suicide attempts. However, no gender comparisons were undertaken [31]. So far, studies investigating gender-related ER differences in clinical cohorts show no relevant gender-specific differences with regard to negative affect, emotional instability, and interpersonal dysfunction in an ED cohort consisting of *n* = 251 females and *n* = 137 males [32] or with regard to emotional overeating in a BED cohort comparing *n* = 172 females and *n* = 48 males [33]. There are also divergent results showing no differences in complex emotion recognition between males with ED (*n* = 29) and healthy controls (HC) (*n* = 42) [34]. However, none of these studies in males made ER-specific instruments. Instead, the studies used subscales from a personality questionnaire as indirect measures to assess both negative affect and interpersonal dysfunction. Others have solely applied a specific measure of overeating in response to emotions, or analyzed only emotion recognition, but not ER strategies or emotion difficulties In addition, no study published to date, to our knowledge, analyzed ER in males using the different DSM-5 ED types, either because the sample size did not allow for it or because they only analyzed one ED type.

Personality traits and ER appear to be intertwined, with evidence showing links between the two in a number of studies [35,36]. For instance, difficulties in ER are implicated in the diagnostic criteria for some personality disorders (e.g., borderline personality disorder) [37]. ED are also associated with specific personality traits, including harm avoidance and low self-directedness in all ED diagnostic types, high novelty seeking in BED and BN, and high reward dependence and persistence in AN [38,39]. Males suffering from ED scored significantly lower than females with ED on harm avoidance, reward dependence, and cooperativeness, had less body image concerns, and lower general psychopathology [40]. In addition, dysfunctional personality traits are associated with higher ED severity, general psychopathology, self-harm behaviors, and worse therapy response and prognosis [41,42,43]. In a previous study by our group, we showed that ER difficulties mediated the relationship between personality traits (i.e., high harm avoidance and low self-directedness) and ED severity [44]. Thus, personality traits may increase vulnerability to ED pathology through ER difficulties. As these aspects were not studied in males with ED before, we incorporated an examination of the interplay between ER, personality traits, ED severity, and ED-related and general psychopathology in males with ED as further objectives of the present study.

Taking into account all the aforementioned gaps in the literature, primarily the lack of studies with clinical samples of males with ED, we aimed to examine ER in a large sample of consecutively recruited male and female patients with ED and HC, considering different DSM-5 ED types. Based on a previous research carried out at our Unit [23], which found how ER strategies improved along with improvements in eating symptoms after cognitive behavioral therapy (CBT), we analyzed the relationship between ED severity, general psychopathology, specific personality traits, and ER. In addition, assessment of the associations between ER and other behaviors commonly used to alleviate aversive emotional states, such as non-suicidal self-injury (NSSI), (reduced) interoceptive awareness, binge-eating, and purging behaviors were part of the study protocol.

## 2. Experimental Section

### 2.1. Participants

The sample consisted of 62 male participants diagnosed with ED (16-AN, 12-BN, 15-BED, 19-Other Specified Feeding or Eating Disorder (OSFED)), 656 female ED patients (140-AN, 236-BN, 100-BED, 180-OSFED), and a HC group, 286 females and 78 males, without a history of ED. The clinical groups were consecutively referred for assessment and treatment at the Eating Disorders Unit within the Department of Psychiatry at Bellvitge University Hospital in Barcelona, Spain. All patients were diagnosed according to the DSM-5 [37] criteria and assessed by senior clinicians specialized in ED. All HC came from the same catchment area as the patients. Participants were recruited between May 2013 and July 2018. In accordance with the Declaration of Helsinki, the present study was approved by the Ethics Committee of our institution (The Clinical Research Ethics Committee (CEIC) of Bellvitge University Hospital). All the participants provided signed informed consent.

### 2.2. Assessment

Eating Disorder Inventory-2 (EDI-2) [45]. This is a reliable and valid 91-item multidimensional self-report questionnaire that assesses different cognitive and behavioral characteristics of eating disorders: Drive for thinness, body dissatisfaction, bulimia, ineffectiveness, perfectionism, interpersonal distrust, interoceptive awareness, maturity fears, asceticism, impulse regulation, and social insecurity. This instrument was validated in a Spanish population [46]. Internal consistency was excellent in our sample (α = 0.97 for the total scale).

Symptom Checklist-90 Items-Revised (SCL-90-R) [47]. This is a 90-item questionnaire that is widely used for assessing self-reported psychological distress and psychopathology. The test is scored on nine primary symptom dimensions: Somatization, obsessive-compulsive, interpersonal sensitivity, depression, anxiety, hostility, phobic anxiety, paranoid ideation, and psychoticism, and three global indices: Global Severity Index (GSI), Positive Symptom Total (PST), and Positive Symptom Distress Index (PSDI). This instrument was validated in a Spanish population [48]. Internal consistency was excellent in our sample (Cronbach’s alpha, α = 0.98 Cronbach’s alpha).

Temperament and Character Inventory–Revised (TCI-R) [49]. The TCI-R is a 240-item questionnaire with a five-point Likert scale format. This questionnaire is a reliable and valid measure of four temperaments (harm avoidance, novelty seeking, reward dependence, and persistence) and three character dimensions (self-directedness, cooperativeness, and self-transcendence). This questionnaire was validated in a Spanish adult population [50]. Cronbach’s alpha for the current sample ranged from good (α = 0.81 for “novelty seeking”) to excellent (α = 0.99 for “persistence”).

Difficulties in Emotion Regulation Scale (DERS) [51]. The DERS assesses emotion dysregulation across six subscales: (a) Nonacceptance of emotional responses, (b) difficulties in pursuing goals when having strong emotions, (c) difficulties controlling impulsive behaviors when experiencing negative emotions, (d) lack of emotional awareness, (e) limited access to emotion regulation strategies, and (f) lack of emotional clarity. Higher scores indicate more difficulties in emotion regulation. The Spanish version was validated in an adult population [44], and excellent internal consistency was found in the study sample (α = 0.96 for the total scale).

### 2.3. Statistical Analysis

Statistical analysis was carried out with Stata15 for Windows. The comparison of quantitative variables between the groups was based on analysis of variance adjusted for the participants’ age, education level, and civil status (ANCOVA). The estimation of the effect size of the pairwise comparisons was based on Cohen’s-d coefficients (|d| > 0.20 was considered low, |d| > 0.5 was considered moderate, and |d| > 0.8 was considered high) [52]. In addition, Finner’s procedure (a familywise error rate stepwise method which has demonstrated more powerful than Bonferroni correction) controlled the increase in Type-I error due to multiple comparisons [53].

Linear multiple regressions stratified by sex estimated the predictive capacity of clinical measures (defined as the independent variables) on ER (defined as the criterion, DERS total score). Each regression was adjusted in five blocks/steps: (a) First block-step entered and set the covariates participants’ age, education, and civil status; (b) Second block added ED-related variables (EDI-2 total, onset of the ED, and duration of the ED); (c) The third block included global psychopathological state (SCL-90R GSI); (d) The fourth block entered NSSI (0 = absent; 1 = present); and (e) The fifth block included personality traits (TCI-R scale scores). The specific predictive capacity of each step/block was measured as the increase in the *R*^2^ coefficient (∆*R*^2^).

Pathways analysis assessed the underlying mechanisms of the following study variables: Participants’ sex and age, personality traits, EDI-2 total score, SCL-90-R GSI and DERS scale scores. This method constitutes an extension of multiple regression modeling, which aims to estimate the magnitude and significance of hypothesized associations in a set of variables with the advantage of allowing for the testing of mediational links (direct and indirect effects) [54]. Structural equation modeling (SEM) was used by defining the maximum-likelihood estimation of parameter estimation and testing goodness-of-fit through standard statistical measures: The root mean square error of approximation (RMSEA), Bentler’s Comparative Fit Index (CFI), the Tucker-Lewis Index (TLI), and the standardized root mean square residual (SRMR). Adequate model fit was considered non-significant by χ^2^ tests and if the following criteria were met [55]: RMSEA < 0.08, TLI > 0.9, CFI > 0.9 and SRMR < 0.1. In this study, ER was defined as a latent variable defined by DERS scale scores, and the personality profile as a latent class defined by TCI-R scale scores.

## 3. Results

### 3.1. Sample Characteristics

Table 1 includes the description and the comparison between the four groups of the study defined by ED diagnosis and sex. Differences emerged with regards to civil status, education and age.

### 3.2. ER and Negative Affect Measures and Comparison between Groups

The first block of Table 2 includes the results of the ANCOVA (adjusted forage, civil status, and education) comparing the four study groups (ED-women, ED-men, HC-women, and HC-men) with regard to DERS scales, EDI-2 scales, and the binge-eating/purging levels (these two last measures were only compared between ED groups). Pairwise comparisons between ED-women and ED-men reached significance in all measures (more ER difficulties for ED-women), except for DERS awareness and the EDI-2 interpersonal distrust (no differences between the two groups were obtained). ED-women registered higher mean scores in all the measures compared to HC-women. The same occurred with ED-men compared to HC-men (except for on EDI-2 perfectionism). No differences between the two HC groups (women and men) were found.

The second block of Table 2 contains the prevalence of NSSI and the comparison between the groups (comparison between the groups was based on logistic regression adjusted by the participants’ age, education, and civil status). The proportion of ED-women who reported the presence of this behavior was higher than the proportion reported by ED-men (44.2% vs. 16.1%, *p* < 0.001), as well as the proportion reported by the HC-women (44.2% vs. 21.8%, *p* < 0.001). No significant differences were found comparing the HC groups (women and men) or between ED-men and HC-men.

Figure 1 includes a radar-chart for the study variables in the four groups. To allow for easy interpretation, z-standardized means were plotted.

### 3.3. Comparison of ER between ED Subtypes

Table 3 includes the ANCOVA (also adjusted for age, education, and civil status) comparing DERS scores between the ED types (AN, BN, BED, and OSFED), stratified by sex. In the female subsample as a whole, greater ER difficulties were associated with BN, followed by BED and OSFED. The lowest DERS scores were found in AN. In the male subsample, greater ER difficulties were registered in OSFED group, followed by the BN and BED groups. AN males had the lowest DERS scores. Results obtained in the men subsample must be interpreted with caution due to the low sample size of the groups.

### 3.4. Predictive Capacity of the Study Variables on ER

Table 4 includes the final models of the two multiple regressions measuring the predictive capacity of study variables on the DERS total score. In the ED-females model, emotion regulation difficulties were predicted by higher EDI-2 total scores, more pronounced psychopathology, higher levels in the novelty seeking and cooperativeness traits, and lower levels in the reward dependence and self-directedness traits. No significant predictive contribution of the NSSI on the DERS-total was found in the ED-females group.

For the ED-males model, DERS-total scores increased for men who reported higher scores on the EDI-2, those with higher psychopathology and lower levels in persistence.

### 3.5. Pathways Analysis

Figure 2 includes the path-diagram with the standardized coefficients of the SEM obtained in the ED group (Appendix A, includes the complete results valuing direct, indirect. and total effects). Goodness-of-fit was obtained (all the fit statistics were in the adequate range). The latent variable measuring ER difficulties (labeled as DERS in the figure) was directly increased for patients who presented higher ED severity (higher EDI-2 total), higher psychopathology (higher SCL-90R GSI), and who were younger. Higher scores in the latent variable measuring the personality construct (labeled as TCI-R in the figure) were also direct predictors of greater ER difficulties. ED severity and the psychopathology levels mediated the relationships between personality measures and ER, as well as between sex and ER: Higher levels in the TCI-R construct and being female increased EDI-2 interoceptive awareness and SCL-90R scores, which contributed to increases on the DERS.

## 4. Discussion

The present study attempted to address a relevant issue in the psychopathology of male patients with ED. It aimed to provide a better knowledge regarding ER in this clinical population, analyzing and comparing ER difficulties between male and female patients with ED and HC, which was rarely studied before. Findings from this study provide new information for the treatment approach of male patients with ED, a minority in the field of ED that runs the risk of being overlooked.

Our first main finding confirmed that patients with ED, both males and females, showed greater global ER difficulties than HC. Although these results are not in accordance with prior research indicating that males with ED did not differ from HC males in emotion regulation strategies, such as emotion recognition [34], they are in line with previous studies which found that negative affect and difficulties in ER predicted disordered eating in both males and females in community samples [26,27]. These discrepancies may be due to the fact that the study by Goddard et al. [34] focused on emotional recognition and not on ER. Moreover, our results support previous findings in clinical samples that have reported decreased effective ER strategies among female patients with ED when compared with HC [10,56], suggesting that a lack of effective ER skills may prompt individuals to use disordered or abnormal eating behaviors to regulate negative affect [57], as well as contribute to body dissatisfaction and disordered eating in males [30]. Therefore, ER difficulties may act as an important etiological feature [57] or risk factor for the occurrence of EDs [23]. Although previous studies have focused primarily on females with ED, our findings also offer the possibility of generalizing these findings to males with ED.

When comparing male and female patients with ED, female patients with ED engaged in more dysfunctional ER strategies than males with ED, displaying greater scores on all DERS scales, except for DERS emotional awareness. There were no differences between male and female controls with regard to ER difficulties. First, these findings confirmed our hypothesis that both males and females with ED displayed a lack of emotional awareness. Second, the fact that female patients with ED scored higher in the most of the DERS scales, such as nonacceptance of emotional responses, limited access to emotion regulation strategies, lack of emotional clarity, and difficulties in engaging in goal-directed behavior or in controlling impulsive behaviors when experiencing negative emotions suggests that there are indeed gender-related patterns of ER in ED. However, we cannot fully exclude gender-related response bias, since males may have had a tendency to minimize or underestimate (intentionally or unintentionally) the difficulties related to their ER in order to prevent their culturally imposed, self-perceived masculinity ideals from being threatened [58]. Furthermore, males with ED appear to more often use externalizing behaviors (e.g., hetero-aggression) or to engage in drug or alcohol use/abuse to deal with emotions whereas females with ED tend to use more internalizing behaviors, such as NSSI [41]. Our results support these observations, with females with ED in the present study showing significantly more NSSI behaviors than males with ED.

Regarding ED types, our findings showed higher ER difficulties in patients with binge eating-related behaviors (BN, BED, and OSFED) compare to patients with restrictive behaviors (AN), in both males and females with ED. These results are consistent with previous studies reporting more ER difficulties among patients with binge-eating behaviors [44,59], but they are discrepant to other studies reporting less severe ER difficulties in BED patients and no significant differences between AN-R and other ED subtypes [5]. However, while females with BN showed the greatest ER deficits compared to females with other ED diagnoses, males diagnosed with OSFED were those who displayed the most ER difficulties. These differences suggest that females and males with ED engage in different disordered eating behaviors for alleviating negative affect and emotional instability. Females with ED seem to present more binge eating and purging behaviors for ER, whereas males with ED are prompt to use more heterogeneous ED-related symptomatology for alleviating emotional distress (e.g., high levels of exercise).

In terms of primary predictors, higher general psychopathology was the shared factor associated with ER difficulties in both males and females with ED. However, ED severity and different personality traits were identified as differential predictors in females and males with ED. Increased ED severity, higher novelty seeking, higher cooperativeness, lower reward dependence, and lower self-directedness were related to higher ER difficulties in females with ED, while lower persistence was associated with ER difficulties in males with ED. Thus, in females with ED, difficulties in ER were associated with a tendency to be more impulsive and intolerant of routine, and which are linked with seeking little emotional support, the unwillingness to be sociable, and having difficulty in expressing feelings and thoughts [60]. On the other hand, in males with ED, difficulties with ER were associated with low persistence, that is, a tendency to be less perseverant in situations of frustration and fatigue [60]. In light of our results, our findings suggest that personality differences may impact ER difficulties, therefore, it would be important to assess for personality traits and consider potential gender-related differences [61,62]. In this regard, it may also be useful to apply ER-based adjuvant treatments focused on reducing impulsivity and increasing self-directedness and reward dependence for females with ED, and specific treatment approaches for males with ED where increased persistence management are specifically addressed.

Finally, another emergent finding was that both ED severity and general psychopathology mediated the relationships between personality and ER difficulties. This may open a new line of research that allows for knowing if the improvement in the ED symptomatology could establish changes in emotion dysregulation. In this sense, a previous study in females with ED found ER improvements after CBT (treatment as usual, without any specific module addressing ER), especially in patients with BN. This study found that improvements in ER were the largest in those with a better treatment outcome [23]. In this line, our results reinforce this concept, suggesting that ED severity and psychopathology may be associated with ER difficulties. In addition, although our study is transversal and does not allow us to analyze the causality, we suggest the existence of a bidirectional pathological process that has ER difficulties acting as a maintenance factors for the ED. However, these findings do not exclude the possibility that ER is also a vulnerability factor for ED. The lack of longitudinal studies analyzing individuals before developing the ED does not allow us to identify if the ER is an etiopathogenic factor of the disorder or if, on the contrary, difficulties in ER are aggravated with the ED. It is most likely that ER is probably acting in both directions, both as a vulnerability factor and as a maintenance factor for the disorder (which is aggravated by psychopathology). With this in mind, we hypothesize that treatment enhanced with a module aimed at improving ER skills could benefit the treatment outcome of ED patients. Further studies should address this point.

Also, the results suggest that, a more dysfunctional personality profile and being female increased the risk of higher ED severity and general psychopathology, which contributed to an increase in ER difficulties in patients with ED. In this vein, a recent study found that depression moderated the association between ER difficulties and binge eating in patients with BED, suggesting that individuals who experience more intense emotions are more affected by difficulties in ER [8]. Again, the above is consistent with the need for treatment based on addressing the difficulties of ER in ED patients, since, although being aware of one’s own emotions is not sufficient for an adaptive emotional regulation [44], it is the first step to improving it.

### Limitations and Strengths

The present study should be evaluated within the context of its limitations. First, as we only assessed patients with ED that were seeking treatment in a clinical setting, the patient cohorts may not be representative of all patients with ED. In addition, ER difficulties were assessed by means of the DERS. Although this is a validated instrument for the assessment of ER, it may not capture other relevant aspects of ER, such as ER strategies or skills (e.g., reappraisal, stimulus control, etc.). Finally, due to the study’s cross-sectional design, no conclusions can be drawn with regard to response to treatment between genders.

Notwithstanding these limitations, the current study has also several strengths that should be noted. One of the strengths of our study includes the relatively large number of males with ED in our sample and our comparison with females with ED, as well as with male and female healthy controls. For the first time, we addressed ER in a large sample of males with ED, including different DSM-5 types. As far as we know, this is the first study assessing predictors of difficulties in ER in females and males with ED.

## 5. Conclusions

There is a growing interest in addressing difficulties in ER in the treatment of patients with ED. However, most ER-based studies were performed in females with ED and, to date, no study was carried out in males with all DSM-5 ED diagnoses. Our findings suggest that treatments focusing on enhancing ER abilities are likely to be beneficial to both female and male patients with ED. Our findings also suggest a bidirectional relationship, that is, if we improve eating symptomatology and general psychopathology, we could improve ER in these patients. However, our results also provide evidence for the need to design specific treatments for males and females with ED that address shared and differential gender-related features associated to emotion dysregulation, such as impulsivity and reward dependence in females, and persistence in males with ED. Taking into account all of the aforementioned factors, further research should be addressed to validate and complement our results, including other measures of ER. Likewise, longitudinal designs may offer insight into gender-related responses of ER difficulties to ED treatments. Findings of this kind may, in fact, provide further evidence for the need of gender-specific, ER-centered treatments as a further step toward individualized psychotherapy.

## Figures and Tables

**Figure 1 jcm-08-00161-f001:**
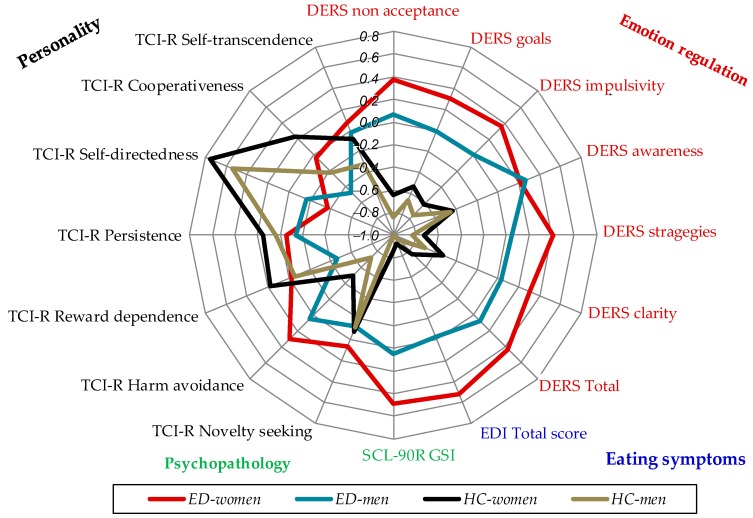
Radar-chart with the z-standardized means by group (*n* = 1082). DERS: Difficulties in Emotion Regulation Scale; ED: Eating disorders; EDI-2: Eating Disorders Inventory-2; HC: Healthy controls; TCI-R, SCL-90R: Symptom Checklist-90 Items-Revised; Temperament and Character Inventory revised.

**Figure 2 jcm-08-00161-f002:**
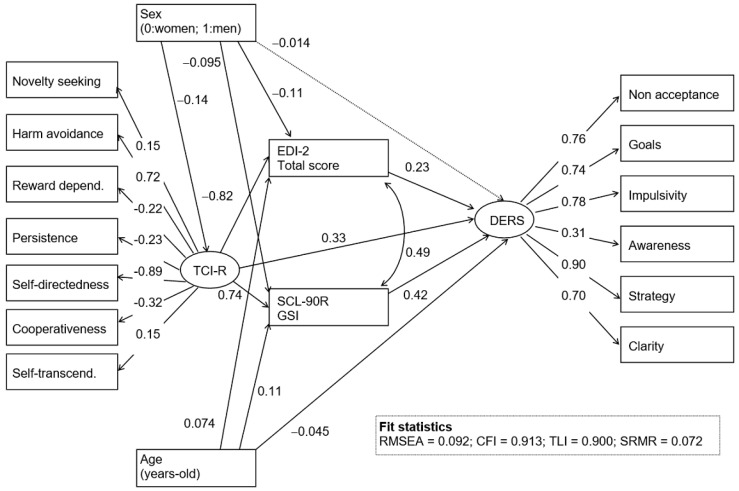
SEM: Standardized coefficients (ED subsample; *n* = 718). Continuous line: Significant parameter (0.05 level). Dash line: Nonsignificant parameter. GSI: Global Severity Index.

**Table 1 jcm-08-00161-t001:** Sample description.

	ED Females		ED Males		HC Females		HC Males		
	(*n* = 656)		(*n* = 62)		(*n* = 286)		(*n* = 78)		
	*n*	*%*	*n*	*%*	*n*	*%*	*n*	*%*	*p*-Value
Civil status									
Single	486	74.1%	42	67.7%	278	97.2%	77	98.7%	**<0.001 ***
Married-partner	114	17.4%	17	27.4%	3	1.0%	0	0.0%	
Separated-divorced	56	8.5%	3	4.8%	5	1.7%	1	1.3%	
Education									
Primary	261	39.8%	28	45.2%	6	2.1%	2	2.6%	**<0.001 ***
Secondary	271	41.3%	22	35.5%	276	96.5%	75	96.2%	
University	124	18.9%	12	19.4%	4	1.4%	1	1.3%	
Employed									
Student	259	39.5%	22	35.5%	120	42.0%	42	53.8%	0.077
Unemployed	397	60.5%	40	64.5%	166	58.0%	36	46.2%	
	Mean	SD	Mean	SD	Mean	SD	Mean	SD	*p*-Value
Age (years-old)	29.78	11.07	33.56	12.73	21.06	4.19	21.30	4.53	**<0.001 ***

SD: Standard deviation. * Bold: Significant comparison (0.05 level). ED: Eating disorder; HC: Healthy control.

**Table 2 jcm-08-00161-t002:** Comparison of DERS scales, EDI-2 scales, and negative affect between groups: ANCOVA adjusted for age, civil status, and education.

	ED Women		ED Men		HC Women		HC Men		ED Women vs. ED Men		HC Women vs. HC Men		ED Women vs. HC Women		ED Men vs. HC Men	
	(*n* = 656)		(*n* = 62)		(*n* = 286)		(*n* = 78)									
	Mean	SD	Mean	SD	Mean	SD	Mean	SD	*p*-Value	|d|	*p*-Value	|d|	*p*-Value	|d|	*p*-Value	|d|
DERS scales																
Non-acceptance	19.63	6.87	17.33	6.68	12.35	5.29	10.99	4.59	**0.007 ***	0.34	0.089	0.27	**<0.001 ***	**1.19 ^†^**	**<0.001 ***	**1.11 ^†^**
Pursuing goals	17.64	5.03	16.07	4.89	12.98	4.12	12.47	4.31	**0.013 ***	0.32	0.402	0.12	**<0.001 ***	**1.01 ^†^**	**<0.001 ***	**0.78 ^†^**
Impulse behaviors	17.00	6.47	14.69	6.23	10.71	4.01	9.89	3.16	**0.003 ***	0.36	0.267	0.23	**<0.001 ***	**1.17 ^†^**	**<0.001 ***	**0.97 ^†^**
Emotional awareness	17.91	5.10	18.25	4.51	14.56	4.12	14.29	4.31	0.594	0.07	0.659	0.06	**<0.001 ***	**0.72 ^†^**	**<0.001 ***	**0.90 ^†^**
Emotional regulation	25.54	8.24	22.39	8.05	15.13	5.87	14.26	5.47	**0.002 ***	0.39	0.370	0.15	**<0.001 ***	**1.45 ^†^**	**<0.001 ***	**1.18 ^†^**
Emotional clarity	14.76	5.07	13.51	5.08	10.22	3.63	9.42	3.67	**0.043 ***	0.25	0.182	0.22	**<0.001 ***	**1.03 ^†^**	**<0.001 ***	**0.92 ^†^**
Total score	112.46	26.94	102.25	26.02	75.95	19.27	71.31	16.95	**0.002 ***	0.39	0.143	0.26	**<0.001 ***	**1.56 ^†^**	**<0.001 ***	**1.41 ^†^**
EDI-2 scales																
Drive for thinness	14.23	6.02	10.51	5.60	3.74	5.04	2.75	3.45	**<0.001 ***	**0.64 ^†^**	0.206	0.23	**<0.001 ***	**1.89 ^†^**	**<0.001 ***	**1.67 ^†^**
Body dissatisfaction	17.22	7.88	11.10	8.32	6.37	6.75	4.63	4.87	**<0.001 ***	**0.75 ^†^**	0.094	0.30	**<0.001 ***	**1.48 ^†^**	**<0.001 ***	**0.95 ^†^**
Interoceptive awareness	11.83	7.06	8.32	6.43	2.83	2.89	1.94	1.99	**<0.001 ***	**0.52 ^†^**	0.249	0.36	**<0.001 ***	**1.67 ^†^**	**<0.001 ***	**1.34 ^†^**
Bulimia	7.21	5.59	3.74	4.15	1.49	1.68	1.01	0.93	**<0.001 ***	**0.70 ^†^**	0.454	0.35	**<0.001 ***	**1.39 ^†^**	**0.001 ***	**0.91 ^†^**
Interpersonal distrust	5.76	4.81	5.67	4.52	2.54	2.92	2.57	2.76	0.880	0.02	0.969	0.01	**<0.001***	**0.81 ^†^**	**<0.001 ***	**0.83 ^†^**
Ineffectiveness	11.90	7.70	8.21	7.22	2.26	3.05	2.02	2.65	**<0.001 ***	**0.51 ^†^**	0.788	0.09	**<0.001 ***	**1.65 ^†^**	**<0.001 ***	**1.14 ^†^**
Maturity fears	8.79	5.97	7.33	5.03	4.46	3.72	4.20	3.38	**0.036 ***	0.27	0.725	0.07	**<0.001 ***	**0.87 ^†^**	**0.001 ***	**0.73 ^†^**
Perfectionism	6.18	4.36	5.05	4.00	3.99	3.55	4.08	3.34	**0.037 ***	0.27	0.874	0.03	**<0.001 ***	**0.55 ^†^**	0.186	0.27
Impulse regulation	6.84	6.03	5.55	5.21	1.28	2.27	1.47	3.06	**0.047 ***	0.23	0.783	0.07	**<0.001 ***	**1.22 ^†^**	**<0.001 ***	**0.95 ^†^**
Ascetism	7.36	4.09	5.81	4.25	2.35	2.22	2.57	2.10	**0.001 ***	0.37	0.653	0.10	**<0.001 ***	**1.52 ^†^**	**<0.001 ***	**0.96 ^†^**
Social insecurity	8.11	5.37	6.56	4.74	2.63	2.85	2.40	2.82	**0.013 ***	0.31	0.725	0.08	**<0.001 ***	**1.28 ^†^**	**<0.001 ***	**1.07 ^†^**
Total score	105.43	42.63	77.88	42.60	33.89	21.52	29.50	15.65	**<0.001 ***	**0.65 ^†^**	0.355	0.23	**<0.001 ***	**2.12 ^†^**	**<0.001 ***	**1.51 ^†^**
Binge eating/purging																
Binge episodes	3.76	6.32	1.86	2.60	---	---	---	---	**0.019 ***	0.39	---	---	---	---	---	---
Purging episodes	3.96	8.75	1.29	3.60	---	---	---	---	**0.018 ***	0.40	---	---	---	---	---	---
	*n*	%	*n*	%	*n*	%	*n*	%	*p*-Value	|d|	*p*-Value	|d|	*p*-Value	|d|	*p*-Value	|d|
^1^ NSSI	290	44.2%	10	16.1%	62	21.8%	19	25.0%	**<0.001 ***	**0.64 ^†^**	0.544	0.08	**<0.001 ***	**0.50 ^†^**	0.260	0.22

SD: Standard deviation; NSSI: Non-suicidal self-injury; ED: Eating disorder; HC: Healthy control; EDI-2: Eating Disorder Inventory-20; DERS: Difficulties in Emotion Regulation Scale. * Bold: Significant comparison (0.05 level); ^†^ Effect size in the moderate (|d| > 0.50) to large range (|d| > 0.80); ^1^ Results obtained in logistic regression. --- Binge and purging episodes were not registered for the HC group.

**Table 3 jcm-08-00161-t003:** Comparison of DERS scales between diagnostic subtypes: ANOVA adjusted for age, civil status, and education.

**Subsample**	**AN**		**BN**		**BED**		**OSFED**		**AN-BN**		**AN-BED**		**AN-OSFED**		**BN-BED**		**BN-OSFED**		**BED-OSFED**	
	***n* = 140**		***n* = 236**		***n* = 100**		***n* = 180**													
Women	Mean	SD	Mean	SD	Mean	SD	Mean	SD	*p*-Value	|d|	*p*-Value	|d|	*p*-Value	|d|	*p*-Value	|d|	*p*-Value	|d|	*p*-Value	|d|
Nonacceptance	17.62	7.26	20.99	6.42	19.36	6.68	19.70	6.89	**<0.001 ***	**0.52 ^†^**	**0.046 ***	0.25	**0.007 ***	0.29	**0.042 ***	0.25	**0.046 ***	0.19	0.700	0.05
Pursuing goals	16.47	5.16	18.55	4.76	17.51	5.11	17.18	5.05	**<0.001 ***	0.42	0.128	0.20	0.209	0.14	0.092	0.21	**0.006 ***	0.28	0.606	0.07
Impulse behavior	15.44	6.97	18.56	6.03	16.51	6.47	16.39	6.26	**<0.001 ***	**0.51 ^†^**	0.225	0.16	0.186	0.14	**0.010 ***	0.33	**0.001 ***	0.35	0.893	0.02
Emot-awareness	17.44	5.27	18.01	4.87	18.58	5.15	17.62	5.24	0.295	0.11	**0.042 ***	0.22	0.749	0.03	0.367	0.11	0.443	0.08	0.155	0.18
Emot-regulation	23.16	8.79	27.29	7.97	25.31	7.75	24.95	7.97	**<0.001 ***	**0.50 ^†^**	**0.041 ***	0.26	**0.049 ***	0.21	**0.049 ***	0.25	**0.004 ***	0.29	0.739	0.05
Emot-clarity	13.77	5.50	15.22	5.00	14.49	4.53	14.77	5.02	**0.008 ***	0.28	0.303	0.14	**0.042 ***	0.19	0.241	0.15	0.371	0.09	0.673	0.06
Total score	103.9	29.7	118.6	24.8	111.7	25.5	110.6	26.3	**<0.001 ***	**0.54 ^†^**	**0.032 ***	0.28	**0.025 ***	0.24	**0.035 ***	0.27	**0.003 ***	0.31	0.752	0.04
**Subsample**	**AN**		**BN**		**BED**		**OSFED**		**AN-BN**		**AN-BED**		**AN-OSFED**		**BN-BED**		**BN-OSFED**		**BED-OSFED**	
	***n* = 16**		***n* = 12**		***n* = 15**		***n* = 19**													
Men	Mean	SD	Mean	SD	Mean	SD	Mean	SD	*p*-Value	|d|	*p*-Value	|d|	*p*-Value	|d|	*p*-Value	|d|	*p*-Value	|d|	*p*-Value	|d|
Nonacceptance	16.14	7.86	17.95	5.65	17.90	7.78	17.79	5.46	0.507	0.26	0.499	0.22	0.494	0.24	0.985	0.01	0.950	0.03	0.965	0.02
Pursuing goals	13.69	4.68	15.83	5.17	16.66	4.52	17.32	4.75	0.266	**0.53 ^†^**	**0.048 ***	**0.65 ^†^**	**0.035 ***	**0.77 ^†^**	0.661	0.17	0.415	0.30	0.711	0.14
Impulse behavior	11.66	4.22	16.11	6.33	14.07	6.24	16.85	6.64	**0.043 ***	**0.83 ^†^**	0.286	**0.51 ^†^**	**0.015 ***	**0.93 ^†^**	0.386	0.32	0.740	0.12	0.203	**0.51 ^†^**
Emot-awareness	17.96	4.75	18.21	3.55	19.28	4.73	17.21	4.81	0.893	0.06	0.455	0.28	0.644	0.16	0.561	0.26	0.570	0.24	0.227	**0.52 ^†^**
Emot-regulation	20.43	9.44	21.45	7.05	21.68	6.53	24.66	8.14	0.737	0.12	0.668	0.15	**0.049 ***	**0.53 ^†^**	0.942	0.03	0.275	**0.52 ^†^**	0.293	**0.50 ^†^**
Emot-clarity	14.31	6.38	12.20	5.19	13.75	4.63	12.71	4.27	0.315	0.36	0.777	0.10	0.387	0.29	0.458	0.31	0.798	0.11	0.591	0.23
Total score	94.2	29.3	101.7	25.1	103.3	24.1	106.5	25.0	0.466	0.28	0.356	0.34	0.180	**0.55 ^†^**	0.878	0.06	0.628	0.19	0.737	0.13

SD: Standard deviation; * Bold: Significant comparison (0.05 level). ^†^ Effect size in the moderate (|d| > 0.50) to high range (|d| > 0.80); AN: Anorexia nervosa; BN: Bulimia nervosa; BED: Binge eating disorder; OSFED: Other specified feeding or eating disorder.

**Table 4 jcm-08-00161-t004:** Predictive model of the DERS total score: Multiple regression stratified by sex (ED subsample, *n* = 718).

	ED Women (*n* = 656)								ED Men (*n* = 62)							
	Coefficients (Model Obtained in the Fifth Block-Step)						Change		Coefficients (Model Obtained in the Fifth Block-Step)						Change	
	B	SE	Beta	*p*-Value	95% CI (B)		Δ*R*^2^	*p*-Value	B	SE	Beta	*p*-Value	95% CI (B)		Δ*R*^2^	B
Covariates							0.009	0.131							0.055	0.395
Age (years-old)	−0.13	0.19	−0.054	0.497	−0.51	0.25			0.11	0.43	0.053	0.809	−0.77	0.98		
Civil status (married)	3.33	1.95	0.055	0.089	−0.50	7.15			−0.51	5.42	−0.009	0.926	−11.45	10.44		
Education level	1.51	0.94	0.041	0.107	−0.33	3.36			2.65	2.43	0.077	0.283	−2.27	7.56		
*ED* variables							0.533	**<0.001 ***							0.488	**<0.001 ***
EDI-2 total	0.19	0.03	0.294	**<0.001 ***	0.13	0.24			0.09	0.07	0.158	0.183	−0.05	0.24		
Onset of ED	−0.02	0.21	−0.005	0.934	−0.42	0.39			−0.17	0.35	−0.076	0.635	−0.88	0.54		
Duration of ED	0.05	0.20	0.017	0.798	−0.34	0.44			−0.37	0.28	−0.155	0.196	−0.94	0.20		
Psychopathology							0.073	**<0.001 ***							0.219	**<0.001 ***
SCL-90R GSI	14.61	1.55	0.397	**<0.001 ***	11.57	17.65			19.39	3.72	0.609	**<0.001 ***	11.88	26.91		
Fourth block/step							0.001	0.946							0.006	0.268
NSSI (0 = no; 1 = yes)	0.20	0.84	0.006	0.809	−1.45	1.86			−0.28	2.97	−0.007	0.924	−6.29	5.72		
*TCI-R*							0.025	**<0.001 ***							0.076	**0.018 ***
Novelty seeking	0.08	0.05	0.048	**0.044 ***	0.01	0.17			0.06	0.15	0.042	0.688	−0.24	0.35		
Harm avoidance	0.00	0.05	−0.004	0.926	−0.10	0.09			0.00	0.17	0.000	0.997	−0.34	0.34		
Reward dependence	−0.09	0.05	−0.054	**0.047 ***	−0.18	−0.01			−0.11	0.15	−0.060	0.470	−0.40	0.19		
Persistence	−0.05	0.04	−0.039	0.174	−0.12	0.02			−0.41	0.14	−0.256	**0.005 ***	−0.69	−0.13		
Self-directedness	−0.25	0.05	−0.197	**<0.001 ***	−0.34	−0.15			−0.21	0.16	−0.189	0.195	−0.54	0.11		
Cooperativeness	0.12	0.05	0.070	**0.023 ***	0.02	0.22			0.13	0.17	0.075	0.456	−0.21	0.47		
Self-transcendence	−0.04	0.05	−0.026	0.335	−0.14	0.05			0.02	0.13	0.012	0.891	−0.25	0.28		

* Bold: Significance parameter (0.05 level). DF: Degrees of freedom. Δ*R*^2^: Increase-change in *R*^2^. ED: Eating disorders; EDI-2: Eating Disorders Inventory-2; NSSI: Non-suicidal self-injury. SCL-90R GSI: Global Severity Index of the questionnaire; Symptom Checklist-90 Items-Revised; TCI-R: Temperament and Character Inventory-Revised; B: Non-standardized B-coefficient; SE: standard error; Beta: Standardized B-coefficient; 95% CI: 95% confidence interval.

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
