# Peer review of "Gender-Related Patterns of Emotion Regulation among Patients with Eating Disorders"

_jcm, 2019, doi:10.3390/jcm8020161_

Reviewer 1 Report

Overall, this study has relevance to the treatment of eating disorders, and its strengths lie in aiming to include an understudied population within the ED treatment literature. However, my primary concerns with this article are related to the rationale provided in the introduction for the study hypotheses, and the theory tested in the study design.

Overall, the authors need to present more of a clearly organized rationale for the proposed theoretical model being tested. The introduction generally explains previous research on ER and EDs, and how there is limited information on how this relates to men with EDs. However, the introduction does not provide sufficient information on why the researchers predict that certain aspects of temperament impact ED severity, greater psychopathology, and in turn, greater emotion regulation difficulties. Furthermore, there is no specific discussion of how ED severity and greater psychopathology would mediate the relationship between temperament and ER. Nor does the theoretical model tested with path analysis necessarily explain or provide rationale for why treatments aimed at improving ER would impact treatment outcomes. With that said, a main question I have is for the structural equation model and the ANCOVA: why is DERS total the dependent variable? Theoretically, in line with previous research in the field, one would assume that the authors are exploring how DERS scores may impact ED pathology, not the other way around.  From a clinical standpoint, this also would appear to be the direction of the relationship of interest, given that the thought is that if one intervenes and targets ER deficits, that would improve outcomes in clinical symptomatology. Second, in the SEM, interoceptive awareness, as measured by the EDI-2, is used as a measure of ED severity. Can you please explain why this is? This is just one aspect of deficits or difficulties seen in EDs, and it makes sense that this one factor would likely impact ER difficulties,however, I don't think it is sufficient to measure the construct of increased ED severity. In sum, I'd like to see these aspects of the study and article addressed prior to publication.

In the Discussion section, on page 17 (lines 325-326) there is mention that “both ED severity and general psychopathology mediated the relationships between personality profile and ER difficulties.” I did not see any mediational test conducted, nor were these results discussed in the results section. Please indicate what method was used to test mediation, and report the values of the indirect effects in the results section. As mentioned previously, it will also be important to explain why this mediation is hypothesized and to discuss potential clinical implications of these findings. 

For the Discussion (Lines 318-324), it seems like a strong conclusion based on the present findings to  imply that it is important to have gender-specific treatment. I think it will be more effective to lighten the language by stating something like "findings suggest that personality differences may impact ER difficulties, and it is important to assess for this and consider potential gender-related differences."

More minor edits:

In the discussion of Table 3 results on page 3, the authors describe  a “worse ER” profile. Please change this wording, as it isn’t clear what is meant by this ( e.g., “greater ER difficulties"). Similarly, with discussion of Table 4 and Figure 2 please adjust the wording used to describe a "worse" profile. Also, please note the table alignment in Table 3 and fix (i.e., the 'd' in 'OSFED' is cut off).

Lastly, please review the article for grammatical mistakes and for readability. At times, sentences seem incomplete. For example, Line 67 on page 9, “In opposition to the affect regulation theory.”

Author Response

January 23, 2019

 Dear Editor,

Thank you for your and the reviewers’ comments. As suggested, we have attached a revised version of the manuscript entitled “Gender-related patterns of emotion regulation among patients with eating disorders" (manuscript ID: jcm-429767) o be re-considered for publication in the Journal of Clinical Medicine.

 We have made changes to the manuscript according to the reviewers’ comments. We have included a 'Revised Manuscript with Track Changes' where 'Track Changes' has been used to indicate where changes have been made (with the sole exception of tables). The manuscript has been prepared according to the journal's instructions. Our answers to the reviewers can be seen below.

 Reviewers' comments:

Reviewer 1:

Overall, this study has relevance to the treatment of eating disorders, and its strengths lie in aiming to include an understudied population within the ED treatment literature. However, my primary concerns with this article are related to the rationale provided in the introduction for the study hypotheses, and the theory tested in the study design.

Overall, the authors need to present more of a clearly organized rationale for the proposed theoretical model being tested. The introduction generally explains previous research on ER and EDs, and how there is limited information on how this relates to men with EDs. However, the introduction does not provide sufficient information on why the researchers predict that certain aspects of temperament impact ED severity, greater psychopathology, and in turn, greater emotion regulation difficulties.

Response: We appreciate the reviewer’s comments. Addressing the reviewer's concerns, we have added information on the rationale for examining inferences between ER, personality traits, ED severity, and psychopathology. For changes please refer to the “introduction” of our manuscript. The changes are as follows:

Personality traits and ER appear to be intertwined, with evidence showing links between the two in a number of studies [35, 36]. For instance, difficulties in ER are implicated in the diagnostic criteria for some personality disorders (e.g., borderline personality disorder) 7 ED have also been associated to specific personality traits including harm avoidance and low self-directedness in all ED diagnostic types, high novelty seeking in BED and BN, and high reward dependence and persistence in AN [38,39]. Males suffering from ED scored significantly lower than females with ED on harm avoidance, reward dependence and cooperativeness, had less body image concerns, and lower general psychopathology [40]. In addition, dysfunctional personality traits have been associated with higher ED severity, general psychopathology, self-harm behaviors, and worse therapy response and prognosis [41-43]. In a previous study by our group, we showed that ER difficulties mediated the relationship between personality traits (i.e., high harm avoidance and low self-directedness) and ED severity [44]. Thus, personality traits may increase vulnerability to ED pathology through ER difficulties. As these aspects have not been studied in males with ED before, we have incorporated an examination of the interplay between ER, personality traits, ED severity, and ED-related and general psychopathology in males with ED as further objectives of the present study.

Furthermore, there is no specific discussion of how ED severity and greater psychopathology would mediate the relationship between temperament and ER. Nor does the theoretical model tested with path analysis necessarily explain or provide rationale for why treatments aimed at improving ER would impact treatment outcomes. With that said, a main question I have is for the structural equation model and the ANCOVA: why is DERS total the dependent variable? Theoretically, in line with previous research in the field, one would assume that the authors are exploring how DERS scores may impact ED pathology, not the other way around.  From a clinical standpoint, this also would appear to be the direction of the relationship of interest, given that the thought is that if one intervenes and targets ER deficits, that would improve outcomes in clinical symptomatology.

Response: We understand the reviewer's concern. Therefore, the following statement has been added in the introduction section: "In addition, a prior study analyzing ER in female ED patients before and after treatment found that emotional dysregulation can be modified as an effect of symptomatological ED improvement [23]. With these controversial results in mind, the question of whether the emotional dysregulation is a vulnerability factor for ED, a factor that maintains and worsens with the ED or both, is still open".

The objectives have also been rewritten to read: "… Based on a previous research carried out at our Unit [23], which found how ER strategies improved along with improvements in eating symptoms after cognitive behavioral therapy (CBT), we analyzed the relationship between ED severity, general psychopathology, specific personality traits and ER"

Also, a rationale clarifying this issue has been added in the discussion section: "... In this sense, a previous study in females with ED found ER improvements after CBT (treatment as usual, without any specific module addressing ER), especially in patients with BN. This study found that improvements in ER were the largest in those with a better treatment outcome [23]. In this line, our results reinforce this concept, suggesting that ED severity and psychopathology may be associated with ER difficulties. In addition, although our study is transversal and does not allow us to analyze the causality, we suggest the existence of a bidirectional pathological process that has ER difficulties acting as a maintenance factor for the ED. However, these findings do not exclude the possibility that ER is also a vulnerability factor for EDs. The lack of longitudinal studies analyzing individuals before developing the ED does not allow us to identify if the ER is an etiopathogenic factor of the disorder or, if, on the contrary, difficulties in ER are aggravated with the ED. It is most likely that ER is probably acting in both directions, both as a vulnerability factor and as a maintenance factor for the disorder (which is aggravated by psychopathology). With this in mind, we hypothesize that treatment enhanced with a module aimed at improving ER skills could benefit the treatment outcome of ED patients. Further studies should address this point".

Second, in the SEM, interoceptive awareness, as measured by the EDI-2, is used as a measure of ED severity. Can you please explain why this is? This is just one aspect of deficits or difficulties seen in EDs, and it makes sense that this one factor would likely impact ER difficulties, however, I don't think it is sufficient to measure the construct of increased ED severity. In sum, I'd like to see these aspects of the study and article addressed prior to publication.

Response: As the reviewer suggested, we have reanalyzed the study considering the total EDI-2 score as a measure of ED severity. Table 2 now contains all the EDI-2 scales, and the multiple predictive models and the pathways analysis use the EDI-2 total score as a measure of ED severity.

In the Discussion section, on page 17 (lines 325-326) there is mention that “both ED severity and general psychopathology mediated the relationships between personality profile and ER difficulties.” I did not see any mediational test conducted, nor were these results discussed in the results section. Please indicate what method was used to test mediation, and report the values of the indirect effects in the results section. As mentioned previously, it will also be important to explain why this mediation is hypothesized and to discuss potential clinical implications of these findings. 

Response: The reviewer is right: it is necessary to use statistical tests to assess both direct and indirect effects. We have included a new table (the new Table S1, supplementary), with the complete results from the SEM, including the significance tests for direct, indirect and total effects. We have included all these new results in a supplementary table. We have chosen to create a supplementary table because it contains a large volume of technical results that could hinder the understanding of the work as a whole if they were included in a normal table.

For the Discussion (Lines 318-324), it seems like a strong conclusion based on the present findings to imply that it is important to have gender-specific treatment. I think it will be more effective to lighten the language by stating something like "findings suggest that personality differences may impact ER difficulties, and it is important to assess for this and consider potential gender-related differences."

Response: We fully agree with the reviewer. Therefore, as the reviewer suggests, the sentence has been changed for the following statement: "In light of our results, our findings suggest that personality differences may impact ER difficulties, therefore, it would be important to assess for personality traits and consider potential gender-related differences [61,62]. In this regard, it may also be useful to apply ER-based adjuvant treatments focused on reducing impulsivity and increasing self-directedness and reward dependence for females with ED, and specific treatment approaches for males with ED where increased persistence management are specifically addressed."

More minor edits:

In the discussion of Table 3 results on page 3, the authors describe a “worse ER” profile. Please change this wording, as it isn’t clear what is meant by this (e.g., “greater ER difficulties"). Similarly, with discussion of Table 4 and Figure 2 please adjust the wording used to describe a "worse" profile.

Response: As suggested by the reviewer, the term "worse ER" has been changed to "greater ER difficulties" throughout the manuscript.

Also, please note the table alignment in Table 3 and fix (i.e., the 'd' in 'OSFED' is cut off).

Response: We have reviewed formats for Table 3, as well as for the remaining tables and figures.

Lastly, please review the article for grammatical mistakes and for readability. At times, sentences seem incomplete. For example, Line 67 on page 9, “In opposition to the affect regulation theory.”

Response: We deeply appreciate the reviewer comments. As suggested, we have attached a revised version of the manuscript that has been revised by a native English speaker, Trevor Steward, to correct grammatical errors and improve the overall quality of the writing. The grammatical changes were not made with "track changes" since it made the reading of the manuscript difficult. But if the reviewer prefers, we can attach another version with all the grammatical changes highlighted.

Reviewer 2 Report

The current study examined gender differences in facets of emotion regulation as well as associations with other indices of psychopathology (e.g., psychological distress) among individuals with and without eating disorders.  The focus on males is important as they are an underrepresented group in the literature on eating disorders. My comments and suggestions for improving the manuscript are listed below.

1.      The authors note that their study involved a “great number of males with ED.” In the introduction, the authors cite some studies that have examined emotion regulation processes in males with eating disorders, and it would be helpful to add how many male participants were included in these prior studies. Although 62 males with EDs may be a lot, it does not seem that way in the context of the number of females that were recruited for the current study (i.e., >600), so perhaps some metric of sample sizes in other studies would be useful for the reader.

2.      Related to the previous comment, I wondered whether a sample size of 62 was actually large enough to find some significant effects. For example, in analyses comparing ER profiles by eating disorder subtype in males. I think it would be important for the authors to include a priori power analyses to ensure that their sample of males with eating disorders was actually sufficient for the analyses that were conducted.

3.      The introduction set up the paper such that I was expecting the focus to be on how emotion regulation (and different facets of emotion regulation) may differentially impact eating disorder symptoms in males vs. females with eating disorders. However, as I understand them, the direction of the analyses is reversed in that emotion regulation is the outcome/dependent variable. More discussion of the directionality of their predictions would be beneficial, including rationale for examining the paths from clinical symptoms to emotion regulation versus the other way as was set up in the introduction.

4.      I think if groups differed by education and relationship status as reported in Table 1, then you might want to include these variables as covariates in subsequent analyses, as I imagine both could impact emotion regulation scores.

5.      In the description of the linear multiple regressions, the authors indicate that the second block included the ED-related variables, namely the EDI-2 interceptive awareness and total score. Why was this particular subscale chosen? If the goal is to examine eating disorder symptoms, it would seem that the drive for thinness, body dissatisfaction, and bulimia subscales would have been used to assess the core cognitive and behavioral features of eating disorders.

Author Response

January 23, 2019

 Dear Editor,

Thank you for your and the reviewers’ comments. As suggested, we have attached a revised version of the manuscript entitled “Gender-related patterns of emotion regulation among patients with eating disorders" (manuscript ID: jcm-429767) o be re-considered for publication in the Journal of Clinical Medicine.

We have made changes to the manuscript according to the reviewers’ comments. We have included a 'Revised Manuscript with Track Changes' where 'Track Changes' has been used to indicate where changes have been made (with the sole exception of tables). The manuscript has been prepared according to the journal's instructions. Our answers to the reviewers can be seen below.

 Reviewers' comments:

Reviewer 2

The current study examined gender differences in facets of emotion regulation as well as associations with other indices of psychopathology (e.g., psychological distress) among individuals with and without eating disorders.  The focus on males is important as they are an underrepresented group in the literature on eating disorders. My comments and suggestions for improving the manuscript are listed below.

1.      The authors note that their study involved a “great number of males with ED.” In the introduction, the authors cite some studies that have examined emotion regulation processes in males with eating disorders, and it would be helpful to add how many male participants were included in these prior studies. Although 62 males with EDs may be a lot, it does not seem that way in the context of the number of females that were recruited for the current study (i.e., >600), so perhaps some metric of sample sizes in other studies would be useful for the reader.

Response 1: Thank you for this helpful comment. We have now added the number of males for all studies on ER in ED cohorts mentioned in the “introduction” part of our manuscript. We have also added information on the limitations of previous studies. The changes are as follows (please refer to “introduction”):

“…Studies investigating gender-related ER differences in clinical cohorts have so far shown no relevant gender-specific differences with regard to negative affect, emotional instability and interpersonal dysfunction in an ED cohort consisting of n = 251 females and n =137 males [32] or with regard to emotional overeating in a BED cohort comparing n = 172 females and n = 48 males [33]. There are also divergent results showing no differences in complex emotion recognition between males with ED (n = 29) and healthy controls (HC) (n = 42) [34]. However, none of these studies in males has made use of ER-specific instruments but instead used subscales from a personality questionnaire as indirect measures to assess both negative affect and interpersonal dysfunction. Others have solely applied a specific measure of overeating in response to emotions or analyzed only emotion recognition, but not ER strategies or emotion difficulties per se. In addition, no study published to date, to our knowledge, has analyzed ER in males using the different DSM-5 ED types, either because the sample size did not allow for it or because they only analyzed one ED type."

2.    Related to the previous comment, I wondered whether a sample size of 62 was actually large enough to find some significant effects. For example, in analyses comparing ER profiles by eating disorder subtype in males. I think it would be important for the authors to include a priori power analyses to ensure that their sample of males with eating disorders was actually sufficient for the analyses that were conducted.

Response 2: The sample size is large enough for most of the statistical analyzes performed in the study, with the exception of the comparisons for eating disorder types for the male subsample (second part of Table 3). It is true that the analyzes in these ANCOVA do not reach adequate statistical power, which means that the p-values obtained in the significance tests have a low capacity to detect true differences between the groups. But it must be kept in mind that we have used both significance tests and coefficients to estimate the effect sizes. Concretely, the ANCOVA procedures reported in Table 3 are completed with Cohen's-d (which has the attributes and advantages of being independent of sample size) as a standardized coefficient to estimate the effect size for the pairwise comparisons.

Still, to address the comment made by the reviewer, we have now added that significance tests in Table 3 in the male subsample should be interpreted with caution due to their low statistical power.

 3.      The introduction set up the paper such that I was expecting the focus to be on how emotion regulation (and different facets of emotion regulation) may differentially impact eating disorder symptoms in males vs. females with eating disorders. However, as I understand them, the direction of the analyses is reversed in that emotion regulation is the outcome/dependent variable. More discussion of the directionality of their predictions would be beneficial, including rationale for examining the paths from clinical symptoms to emotion regulation versus the other way as was set up in the introduction.

Response 3: As the reviewer suggested, the following rationale has been added in the manuscript, in the introduction: "In addition, a prior study analyzing ER in female ED patients before and after treatment found that emotional dysregulation can be modified as an effect of symptomatological ED improvement [23]. With these controversial results in mind, the question of whether the emotional dysregulation is a vulnerability factor for ED, a factor that maintains and worsens with the ED or both, is still open".

The objectives were also rewritten to clarify this concern: "… Based on a previous research carried out at our Unit [23], which found how ER strategies improved along with improvements in eating symptoms after cognitive behavioral therapy (CBT), we analyzed the relationship between ED severity, general psychopathology, specific personality traits and ER"

Finally, the following has also been added to the discussion section: "... In this sense, a previous study in females with ED found ER improvements after CBT (treatment as usual, without any specific module addressing ER), especially in patients with BN. This study found that improvements in ER were the largest in those with a better treatment outcome [23]. In this line, our results reinforce this concept, suggesting that ED severity and psychopathology may be associated with ER difficulties. In addition, although our study is transversal and does not allow us to analyze the causality, we suggest the existence of a bidirectional pathological process that has ER difficulties acting as a maintenance factors for the ED. However, these findings do not exclude the possibility that ER is also a vulnerability factor for EDs. The lack of longitudinal studies analyzing individuals before developing the ED does not allow us to identify if the ER is an etiopathogenic factor of the disorder or, if, on the contrary, difficulties in ER are aggravated with the ED. It is most likely that ER is probably acting in both directions, both as a vulnerability factor and as a maintenance factor for the disorder (which is aggravated by psychopathology). With this in mind, we hypothesize that treatment enhanced with a module aimed at improving ER skills could benefit the treatment outcome of ED patients. Further studies should address this point".

4.      I think if groups differed by education and relationship status as reported in Table 1, then you might want to include these variables as covariates in subsequent analyses, as I imagine both could impact emotion regulation scores.

Response 4: We have reviewed all the analyses including education and civil status as new covariates in the comparative and predictive analyses included in Tables 2-3-4. The new results are very similar to the previous ones, most likely because the new variables included as control measures had a strong association with the covariate used in the previous version (age).

5.      In the description of the linear multiple regressions, the authors indicate that the second block included the ED-related variables, namely the EDI-2 interceptive awareness and total score. Why was this particular subscale chosen? If the goal is to examine eating disorder symptoms, it would seem that the drive for thinness, body dissatisfaction, and bulimia subscales would have been used to assess the core cognitive and behavioral features of eating disorders.

Response 5: In the revised version of the article, we have added all the EDI-2 scales in Table 2, since, as the reviewer indicates, it is relevant in knowing the differences between the study groups in all these measures. In the regression model, adding an extensive number of scales for the EDI-2 along with the total score is not possible for two reasons: a) the resulting equation was very extensive and did not fit properly; b) the total score is generated as a linear combination of the subscales, which leading to the problem of collinearity. We have chosen to keep only the total EDI-2 in the regression model. The new results are in the revised version of Table 4 and in the text. The pathways analyses also have changed the EDI-2 interoceptive awareness subscale for the EDI-2 total score.

Round  2

Reviewer 1 Report

Thank you for addressing my previous comments and concerns. This revision appears to adequately address them.

Reviewer 2 Report

I think the authors adequately responded to the reviewer's comments.